# T-Cell Assay after COVID-19 Vaccination Could Be a Useful Tool? A Pilot Study on Interferon-Gamma Release Assay in Healthcare Workers

**DOI:** 10.3390/diseases10030049

**Published:** 2022-07-31

**Authors:** Silva Seraceni, Elena Zocca, Tamara Emanuela Cervone, Flaminia Tomassetti, Isabella Polidori, Massimiliano Valisi, Francesco Broccolo, Graziella Calugi, Sergio Bernardini, Massimo Pieri

**Affiliations:** 1Lifebrain RDI srl, Limena, 35010 Padua, Italy; silva.seraceni@lifebrain.it (S.S.); elena.zocca@lifebrain.it (E.Z.); tamaraemanuela.cervone@lifebrain.it (T.E.C.); 2Department of Experimental Medicine, University of Rome “Tor Vergata”, 00133 Rome, Italy; flaminia.tomassetti@students.uniroma2.eu (F.T.); bernards@uniroma2.it (S.B.); 3Lifebrain srl, Guidonia Montecelio, 00133 Rome, Italy; isabella.polidori92@gmail.com (I.P.); graziella.calugi@laboratoriogenoma.it (G.C.); 4Cerba HealthCare Italia, 20139 Milan, Italy; massimiliano.valisi@cerbahealthcare.it (M.V.); francesco.broccolo@cerbahealthcare.it (F.B.); 5Department of Medicine and Surgery, University of Milan-Bicocca, 20900 Monza, Italy; 6Department of Laboratory Medicine, Tor Vergata University Hospital, 00100 Rome, Italy

**Keywords:** adaptive responses, COVID-19, IGRA test, INF-γ, T-cells, vaccination

## Abstract

*Background:* SARS-CoV-2 T-cells are crucial for long-term protection against reinfection. The aim was to demonstrate the Interferon-gamma Release Assay (IGRA) test could be useful for vaccination monitoring. *Methods:* In a prospective cohort of 98 vaccinated healthcare workers for SARS-CoV-2, we selected 23 people in low-antibodies (Group 1, *N* = 8), high-antibodies (Group 2, *N* = 9), and negative control groups (Group 3, *N* = 6). SARS-CoV-2-specific humoral and cellular responses were analyzed at 8 months after two doses of Pfizer BioNTech, evaluating anti-RBD (Receptor Binding Domain) and RBD-ACE2 (Angiotensin Converting Enzyme-2) blocking antibodies in sera through a Chemiluminescence Immunoassay (CLIA) and T-cells through the IGRA test in heparinized plasma. Moreover, lymphocyte subtyping was executed by a flow cytometer. Statistical analysis was performed. *Results:* The data confirmed that RBD and RBD-ACE2 blocking ACE2 antibody levels of Group 1 were significantly lower than Group 2; *p* < 0.001. However, T-cells showed no significant difference between Group 1 and Group 2. *Conclusions:* This work suggests the need for new strategies for booster doses administration.

## 1. Introduction

As is known, the immunological response is defined by innate and adaptive immunity, where T-cells and B-cells play a leading role [1]. The specific T-cell response is directly related to understanding how humoral and cellular responses of adaptive immunity are activated and work in immunological protection [2]. In the COVID-19 (Coronavirus Disease 2019) pandemic era, growing evidence has also recognized the importance of cellular responses to SARS-CoV-2 (Severe Acute Respiratory Syndrome Coronavirus 2) infection or vaccination [3]. According to recent research, SARS-CoV-2 specific memory T-cells are believed to be crucial for long-term immune protection against COVID-19. Nevertheless, SARS-CoV-2 T-cell-specific responses may also have diagnostic value, as antibody levels have been reported to decline faster than T-cells [4,5]. Some works have affirmed a correlation between an adequate cell-mediated response (INFγ release) based on the severity of the disease, which could be useful to determine adequate SARS-CoV-2 future booster doses [6,7]. However, limited research has been conducted on the T-cell immune response, particularly after vaccination [8,9].

Until this point, millions of serological tests are available against SARS-CoV-2 antibodies, usually detecting anti-RBD (Receptor Binding Domain) or ACE2 (Angiotensin Converting Enzymes-2) competitive antibodies [10,11]. However, ELISA assays are being developed for the indirect quantification of T-cell memory activity, recently [12]. Activated T-cells are characterized by specific peptides that can serve as diagnostic markers. One of the most important markers that is secreted is Interferon-γ (IFNγ) [13].

Using an automated, easy-to-use whole-blood interferon-gamma release assay (IGRA), we demonstrated that most vaccinated individuals have a robust T- and B-cell reactivity against multiple structural proteins, detected in peripheral blood, that may serve as a useful diagnostic tool in managing the COVID-19 pandemic.

## 2. Materials and Methods

Serological antibodies monitoring was performed on 98 health workers, 79 women (mean age 32 ± 12 years, y) and 19 men (mean age 47 ± 10 y), from Lifebrain laboratory for a period ranging from vaccine administration (Pfizer BioNTech, Pfizer, New York, NY, USA) up to 8 months later. During the period, all the healthcare workers were evaluated for SARRS-CoV-2 through molecular analysis to assess the negativity. In this window of time, 10 out of 98 were discovered positive for SARS-CoV-2 swabs and were excluded from the study. Other exclusion criteria were dropouts (8 out of 98), pharmacological therapy (1 out of 98), and booster dose administration in COVID-19 convalescent subjects (2 out of 98). Among the 77 healthcare workers, the RBD antibody trend was evaluated throughout this period; it was noted that some people had high antibody levels, while others had low antibody levels (Appendix A). Subjects showing a profile with an upward trend at the beginning and a moderate downward slope after months were included, while subjects with a fluctuating trend were excluded as described in Table 1. Therefore, it was decided to deep investigate these individuals with a humoral immune response that was reduced or absent and the ones with higher antibody levels that persisted over time: 23 out of 98, 17 women (mean age 44 ± 12 y) and 6 men (mean age 50 ± 10 y). The 23 individuals were divided into Group 1 (No. 8 out of 23), low antibody levels people; Group 2 (No. 9 out of 23), high antibody levels people; and Group 3, negative control group (No. 3 out of 23), people not previously infected and no vaccination. 

Serum samples were collected using tubes containing separating gel, centrifuged, and used freshly for antibody assay. The sera were collected at 8 months after the second dose for Group 1 and Group 2 and after the negative PCR for Group 3. Lithium heparin whole blood was collected after 8 months and was used for the IGRA assay, as described by the manufacturer. The current study was conducted in accordance with the Helsinki Declaration and all subjects that were recruited for the study expressed informed consent for being enrolled. The study was approved by the Ethical Committee of the University of Rome “Tor Vergata”, (approval number: R.S.44.20).

The SARS-CoV-2 cellular response was measured using a specific quantitative interferon-γ release assay in whole blood following the manufacturer’s instructions (SARS-CoV-2 IGRA stimulation tube set, Euroimmun, Lübeck, Germany). Briefly, lithium heparinized blood from each patient was incubated for 21 h at 37 °C in the three tubes that were supplied. Then, the samples were centrifugated at 1300× *g* for 10 min, and the plasma was frozen. In the test procedure, the T-cells in the patient samples are stimulated using spike protein-based antigens in the provided tubes, and the released IFN-γ is subsequently measured using a fully automated quantitative ELISA as per the manufacturer’s instructions (Quant-T-Cell Sars-CoV-2 & Quan-T-Cell ELISA, EUROIMMUN, Perkin Elmer Company). The method cut-off is set at 200 mIU/mL. The upper limit of quantification achieved was 5000 mIU/mL.

The Mindray SARS-CoV-2 S-RBD IgG (Mindray S-RBD IgG) is a two-step CLIA for the quantitative determination of SARS-CoV-2 S-RBD IgG in human serum or plasma, that is performed on the fully automated Mindray CL 1200i analytical system (Mindray Bio-Medical Electronic Co Ltd., Shenzhen, China). The cut-off value in BAU/mL and the linearity range in BAU/mL are respectively: 12.16, 3.65–1216 BAU/mL, as declared by the manufacturer. Samples with values over 1216 BAU/mL were diluted and measured 1:10, allowing an extension of the dynamic range of analysis to 12,160 BAU/mL.

The “SARS-CoV-2 Neutralizing Antibody” CLIA (Mindray Medical, Shenzhen, China) for the detection of anti-SARS-CoV-2 ACE2RBD-ACE2 blocking antibodies was performed on the fully automated Mindray CL-1200i analytical system (Mindray Medical, Shenzhen, China). Neutralizing antibodies in the sample competes with ACE2-alkaline phosphatase conjugate for binding sites of SARS-CoV-2 antigens. As declared by the manufacturer, the conversion factor to transform AU/mL into IU/mL is 4.33 IU/mL. Samples with values over 1732 IU/mL were diluted and measured at 1:10, allowing the extension of the dynamic range of analysis. The linearity range is 8.66 IU/mL to 1732 IU/mL, the cut-off value is 43.3 IU/mL. The sensitivity and specificity are 95.7% and 99.9%, respectively.

Lymphocyte subtyping has been performed with a flow cytometer Beckman Coulter Navios EX provided with optic and laser technology. Single typing has been detected by specific reagent Beckman Coulter as follows CD45-FITC/CD4-RD1/CD8-ECD/CD3-PC5 (CYTO-STAT tetraCHROME), CD45-FITC/CD56-RD1/CD19-ECD/CD3-PC5 (CYTO-STAT tetraCHROME), and A07766 (CD16-PE).

In the case of normally distributed data, descriptive analyses were represented by the mean ± standard deviation and ANOVA with the Bonferroni post hoc test to determine the differences between the groups, otherwise, if only two groups were present, the Student’s t-test was used. In the case of not normally distributed data, they were represented by the median and the percentiles; the variables were compared through the Kruskal–Wallis test for more than two groups or Mann–Whitney test for two groups. A Shapiro–Wilk’s test was used for testing the normality of data with a confidence interval (CI) of 95%. All data were examined using Med Calc Ver.18.2.18 (MedCalc Software Ltd., Ostend, Belgium).

## 3. Results

It was decided to deeply investigate 23 health workers out of 98 with an upward trend at the beginning and a moderate downward slope. Particularly, we investigated the individuals with a humoral immune response that reduced after a SARS-CoV-2 vaccine administration and the ones with an overproduction of antibodies.

### 3.1. Persistence of Antibody Levels

Figure 1 showed the persistence at 8 months after the second dose administration of antibody anti-SARS-CoV-2 levels in Group 1, which had lower antibody levels; Group 2, which had higher antibody levels; and Group 3, the negative control. Figure 1 A confirms that the median concentration value of RBD antibody levels of Group 1 (median value 28.898 BAU/mL, interquartile range, IQR: 27.445 to 33.969 BAU/mL; *N* = 8) is significantly lower than the median of Group 2 (median value 645.295 BAU/mL, IQR: 376.343 to 1072.254 BAU/mL; *N* = 9) with a *p* < 0.001; the median of Group 3 (0.562 BAU/mL; IQR: 0.00 to 2.736 BAU/mL; *N* = 6) is significantly different from Group 1 (*p* < 0.01) and Group 2 (*p* < 0.001). Figure 1B shows the significant difference of ACE2RBD-ACE2 blocking antibody levels between Group 1 (median value 56.550 IU/mL, IQR: 40.745 to 67.743 IU/mL; *N* = 8) and Group 2 (median value 836.816 IU/mL, IQR: 785.159 to 1198.728 IU/mL; *N* = 9) with a *p* < 0.001; the Group 3 (median value 2.16 IU/mL; IQR: 0.00 to 11.149 IU/mL; *N* = 6) is significantly different from Group 1 (*p* < 0.01) and Group 2 (*p* < 0.001).

### 3.2. T-Cells between the 3 Groups

However, despite the heavy differences in the antibody levels, Figure 2 highlighted no significance in T-cells that were measured by the IGRA test (*p*: no significance, n.s.). The median concentrations were, respectively, 633 mLU/mL (IQR: 367.830 to 1525.385) mLU/mL for Group 1 and 1791.650 mLU/mL (IQR: 1025.893 to 1935) mLU/mL for Group 2. Group 3 (median 0.5 mLU/mL; IQR: 0.5 to 5.652 mLU/mL) is significantly different from both groups (*p* < 0.001).

### 3.3. Lymphocyte Subtyping

In the 23 health workers that were investigated, to verify each different cell-mediated response lymphocyte subtyping was performed. Figure 3A is an illustration of the lymphocyte subtyping that was analyzed in this study. Following the Mann–Whitman test for non-parametric data, we evaluated no significant differences between Group 1 and Group 2 in cell populations. In Figure 3B, the median, minimum and maximum values, and the IQR are reported for Group 1 and Group 2 (*N* = 17).

## 4. Discussion

Several studies have been driving to consider T-cell monitoring through the IGRA test, to accurately assess SARS-CoV-2 immunity and potentially influence decision-making for the vaccine schedule [14]. The increasing evidence of the T-cells’ usefulness is also related to their responses that are elicited by SARS-CoV-2 vaccination or natural infection [5]. Our results supported the actual role of SARS-CoV-2 T-cell, which have diagnostic value, being widely expressed after the vaccine administration and persisting at high concentrations over time (after 8 months from the second dose), despite the humoral response [4,15].

Moreover, since the IGRA test is accurate and fully automated, it can be easily performed in clinical laboratories [14] and might be used to quantify T-cell response. Even though the assay requires a 24-h sample set-up that is long and time-consuming, the IGRA test is already known in routine laboratory practice since the same method is used for the diagnosis of tuberculosis, known as Quantiferon. Thus, the IGRA test seems to be useful to verify the cellular immune response in the population that was studied, particularly in the case of a humoral immune response that is reduced after a SARS-CoV-2 after vaccination against this virus (Group 1). Furthermore, it would be used for evaluating the immune response of immunosuppressed or oncological patients or in individuals, who have congenital or acquired immunodeficiencies, such as HIV infection, autoimmune diseases, etc.

After the Pfizer BioNTech vaccine administration in healthcare workers, it was observed that IFNγ concentrations, such as T-cells, showed for Group 2 a higher trend compared to Group 1 but lacked statistical significance, probably due to the variability of the data, despite the significant difference (*p* < 0.001) between the antibody levels in the two groups that were considered. 

A robust T-cell immune response that was elicited after a double-dose vaccine administration, appeared to be not antibody-related, despite other findings, in which SARS-CoV-2 IGRA agreed with levels of anti-Spike antibodies [12,16]. Nevertheless, any significant difference was noted in lymphocyte subtyping between the two groups, showing no alterations in values of CD3+ CD4+ or CD3+ CD8+ T-cells. These data would confirm the IGRA test analysis.

However, consistently with recently published data, it was noted antibody kinetics rapidly decay instead of T-cell concentrations, which persist over time [9]. Therefore, the IGRA test assesses a more complete perspective of the immune response after a SARS-CoV-2 infection or vaccination, particularly when the antibody levels are expressed in low levels, and it should be associated with the serological antibody assays to evaluate the protection rate and to decide how to manage the vaccine schedule [17], especially in the highest COVID-19 risk-exposed individuals, such as healthcare workers [18]. Moreover, cellular immunity to SARS-CoV-2 variants has been shown to persist even when virus cells escape humoral immunity [19,20]. This strengthens the application of the IGRA test in addition to antibody monitoring to the clinical laboratory activities for controlling the variants of concern (VOCs) of SARS-CoV-2 overspread in the population.

Nevertheless, this work is a small pilot study, and the sample size should be increased to establish the effective role of the IGRA test in the clinical area. Unluckily, when the testing was started, the kit availability was truly limited, therefore, just a few significant healthcare workers out of the 98 were selected as participants of the study. However, these findings should be confirmed in larger cohorts. It would be interesting to understand whether the SARS-CoV-2 immunoreactivity and T-cell stimulation improve after the third booster dose, or if the increase is antibody-related only.

## 5. Conclusions

This work provides further insight into the immune response after COVID-19 vaccination, underlining the need to define and optimize vaccine administration and future additional doses. In conclusion, the IGRA test proved to be a useful tool and could be associated with an antibody assay to determine the SARS-CoV-2 immunity in a clinical laboratory. Since T-cell concentrations, unlike antibody levels, remain stable over time, boosting the population over and over with the same vaccine formula seems to be unnecessary.

## Figures and Tables

**Figure 1 diseases-10-00049-f001:**
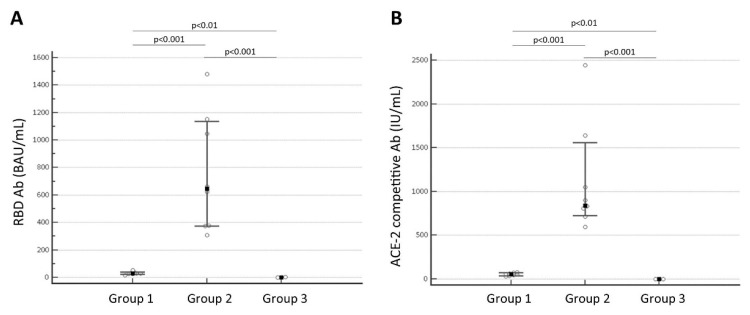
(**A**) The RBD antibodies levels of the three groups that were analyzed (Group 1 median value 28.898 BAU/mL, Group 2 median value 645.295 BAU/mL, Group 3 median value 0.562 BAU/mL). There was a significant difference between Group 1 and Group 2 (*p* < 0.001), between Group 1 and Group 3 (*p* < 0.01), and between Group 2 and Group 3 (*p* < 0.001). (**B**) The ACE2RBD-ACE2 blocking antibody levels of the three groups that were analyzed (Group 1 median value 56.550 IU/mL, Group 2 median value 836.816 IU/mL, Group 3 median value 2.16 IU/mL). There was a significant difference between Group 1 and Group 2 (*p* < 0.001), between Group 1 and Group 3 (*p* < 0.01), and between Group 2 and Group 3 (*p* < 0.001).

**Figure 2 diseases-10-00049-f002:**
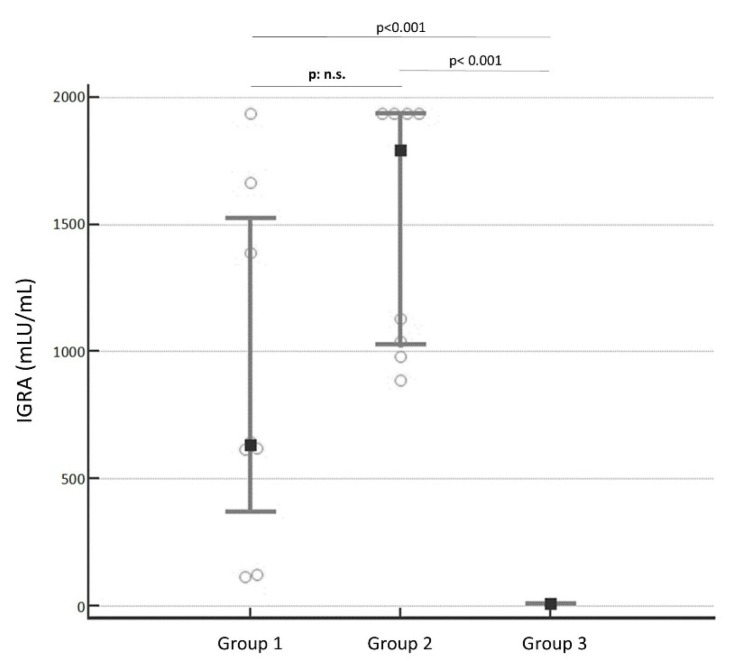
The IGRA concentration between the three groups. It was no noted significant difference between Group 1 and Group 2 (*p*: not significant, n.s.). Group 3 is significantly different from Group 1 (*p* < 0.001) and Group 2 (*p* < 0.001).

**Figure 3 diseases-10-00049-f003:**
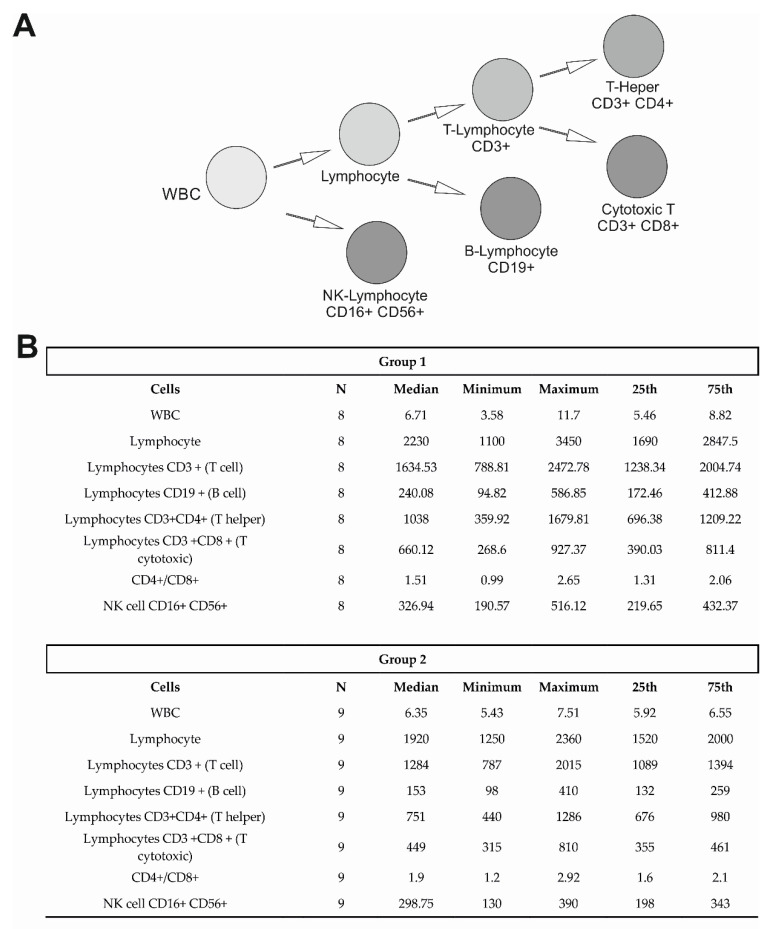
(**A**) The lymphocyte population that was examined. (**B**) Lymphocyte populations that were analyzed. The table reports the median, the interquartile range (25th and 75th percentiles), and the minimum and maximum values. It was observed no significant difference in all the lymphocyte classes. (WBC: white blood cells; NK: Natural Killer; T: thymocytes; B: Bone marrow cells).

**Table 1 diseases-10-00049-t001:** Inclusion and exclusion criteria of the subjects that were enrolled for the IGRA test (No. 23 out of 98).

Inclusion Criteria	Exclusion Criteria
Group 1: subjects with RBD Ab levels constantly low during the monitoring	Subjects with fluctuating antibody trends during the 8 month-period monitoring
Group 2: subjects with RBD Ab levels that persisted at >1000 BAU/mL after months	Subjects who have been positive during the monitoring
Group 3: subjects with no previous COVID-19 infection or vaccination	Subjects under anti-inflammatory therapy
	Dropped out during the monitoring
	Subjects undergone booster dose after COVID-19 infection

## Data Availability

Not applicable.

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
