# Peer review of "T-Cell Assay after COVID-19 Vaccination Could Be a Useful Tool? A Pilot Study on Interferon-Gamma Release Assay in Healthcare Workers"

_diseases, 2022, doi:10.3390/diseases10030049_

Round 1

Reviewer 1 Report

This study evaluated the COVID-19 specific antibody titers as well as the status of T cells activation from 98 health works after 8 months vaccination. Their data offer important information regarding the long-term vaccine induced-immune status. However, there are still some questions that need further clarification.

Major:

1.     In Abstract, please describe the definition of group 1, 2 and 3.

2.     The term "ACE-2 competitive antibody" is easy misunderstanding, it should be revised to SASR-CoV-2 neutralizing antibody or RBD-ACE2 blocking antibody.

3.     The results showing the antibody titers in either IU/mL or BAU/mL after 8 months vaccination were a little lower than other recent publications  (such as Vaccines, 2022 Mar 24;10(4):506). Please clarify it and discuss the difference or influence factors.

4.     The INF-r detected in group 2 showed a higher trend compared to group 1 but lacking statistical significance. Please discuss the possible reasons.    

Minor:

1.     The ethical statement is missing.

2.     The resolution of each figure is too low, please enhance the dpi to at least 300.

Author Response

Comments from Reviewer #1:

Major revision:

Comment 1: In Abstract, please describe the definition of group 1, 2 and 3.

Response: Thank to the reviewer and we included this information in the Abstract

Comment 2: The term "ACE-2 competitive antibody" is easy misunderstanding, it should be revised to SASR-CoV-2 neutralizing antibody or RBD-ACE2 blocking antibody.

Response: Thank the reviewer for pointing this out. We agree with this comment; therefore, we have changed the terminology into RBD-ACE2 blocking antibody.

Comment 3: The results showing the antibody titers in either IU/mL or BAU/mL after 8 months vaccination were a little lower than other recent publications (such as Vaccines, 2022 Mar 24;10(4):506). Please clarify it and discuss the difference or influence factors.

Response: Thanks to the reviewer for this comment, however, the quantification of antibody levels is kit related. In particular, the study suggested used an ELISA assay to determine anti-SARS-CoV-2 IgG levels. Specifically, as declared by the manufactory (EUROIMMUN, Lübeck, Germany), the assay detects IgG anti Domain S1 in the viral Spike protein, while the methods used in our study were two Chemiluminescence assays (CLIA), which detect anti-S-RBD (Receptor Binding Domain) IgG and competitive antibody against ACE2. The second-last method is against a particular region of S1 protein, as well, the last method inhibits the virus access through competition with ACE2.  In another study, we also evaluated the correlation between these two methods and in vitro neutralization (Cristiano A. et al. 2022, doi:10.1016/j.clim.2021.108918). Therefore, the antibody detected in our study have shown a different and deeper sight in the immunological response.

Comment 4: The INF-r detected in group 2 showed a higher trend compared to group 1 but lacking statistical significance. Please discuss the possible reasons.   

Response: Thanks for the suggestion; the two median values are different but due to the variability of the data, they have very high percentiles that do not determine a significant difference between groups 1 and 2. Furthermore, 4 samples of group 2 report the maximum value obtained from the test, probably they could have higher values, but with the data obtained the statistical test detects a p <0.05. We have thus changed a sentence in the discussion about it.

Minor points:

Comment 1: The ethical statement is missing.

Response: The ethical statement has been added in the paragraph Methods.

Comment 2: The resolution of each figure is too low, please enhance the dpi to at least 300.

Response: We thank the reviewer, and we are sorry for the bad resolution, however, each figure uploaded has 600 dpi. Maybe the pdf generated do not preserve the resolution; anyhow we will upload the figure again in the system.

Reviewer 2 Report

The review article entitled “T-Cell Assay after COVID-19 vaccination could be a useful tool? A pilot study on Interferon-gamma Release Assay in healthcare workers” cover T-cell Assay after COVID-19 vaccination, and the authors tried to give a possible answer to the question that if booster dose is needed or not. The brief report seems quite interesting; however, some vital information is missing in the manuscript as follows:

The article needs major English language revision. Many sentence structures are vague, and the scientific writing style is not well kept. There are many typos; for example, in Line 181, “despite other fundings à findings” So, it is recommended that the article be revised for English editing.

1.       There are many vague sentences; for example Line 22, “SARS-CoV-2 T cells are crucial for long-term protection.” Doesn’t address protection against what? Similarly, in lines 61-63, “Throughout this period, it was noted that some people had a high antibody levels trend, while others had a low levels trend (undiscussed data). Therefore, it was decided to deep investigate these 23 individuals” Does not clearly address why 23 people were chosen, and the remaining 75 don’t.

2.       The authors first selected 98 people as a part of the study, but later, they chose the Cohort of 23 people only, which is too tiny to address the cause significantly. Thus, results can be biased. Please provide details about the selection criteria.

3.       The discussion part is too short, and overall citations are 17 only. Both can be further improved. So, please consider re-writing the discussion part.

4.       More supplementary data may be needed to qualify the claims in this manuscript. Thus, it would be appreciated if data from all 98 patients were provided as supplementary.

5.       The conclusion is also unclear in the title authors want to assess T-cell Assay after Covid-19, but the conclusion doesn’t provide any insights into how T-cell Asay can be interpreted for Covid-19 vaccination? What was the exact conclusion of the study, and how can it be utilized? Please consider revising it again.

Author Response

Comments from Reviewer #2:

The review article entitled “T-Cell Assay after COVID-19 vaccination could be a useful tool? A pilot study on Interferon-gamma Release Assay in healthcare workers” cover T-cell Assay after COVID-19 vaccination, and the authors tried to give a possible answer to the question that if booster dose is needed or not. The brief report seems quite interesting; however, some vital information is missing in the manuscript as follows:

The article needs major English language revision. Many sentence structures are vague, and the scientific writing style is not well kept. There are many typos; for example, in Line 181, “despite other fundings à findings” So, it is recommended that the article be revised for English editing.

Comment 1: There are many vague sentences; for example Line 22, “SARS-CoV-2 T cells are crucial for long-term protection.” Doesn’t address protection against what? Similarly, in lines 61-63, “Throughout this period, it was noted that some people had a high antibody levels trend, while others had a low levels trend (undiscussed data). Therefore, it was decided to deep investigate these 23 individuals” Does not clearly address why 23 people were chosen, and the remaining 75 don’t.

Response: We thank the reviewer for the suggestion, and we are sorry for the presence of typo errors; we corrected them, and we included the missing information. Furthermore, we added in the paragraph of Material and Methods the inclusion and exclusion criteria, as Table 1, detailing the choice of testing just 23 people.

Comment 2: The authors first selected 98 people as a part of the study, but later, they chose the Cohort of 23 people only, which is too tiny to address the cause significantly. Thus, results can be biased. Please provide details about the selection criteria.

Response: Thank the reviewer for pointing this out. As described in the Inclusion/Exclusion criteria, the IGRA test chosen to verify the cellular immune response was at the time a new assay just launched on the market with limited distribution. For this, we have been led to choose between our cohort of subjects. Therefore, we decided to investigate particularly the individuals with a humoral immune response reduced after a SARS-CoV-2 vaccine administration and the ones with an overproduction of antibodies. In this way, we could capture the two ends of our populations. Also, we clarified in the text.

Comment 3: The discussion part is too short, and overall citations are 17 only. Both can be further improved. So, please consider re-writing the discussion part.

Response: Thank the reviewer for her/his comment and we improved the discussion, arguing our thesis: lines 189-195, 203-208. Also, we tried to amplify the references, however, it is to consider that the cellular response after vaccination in an open field, with no published research still going on.

Comment 4: More supplementary data may be needed to qualify the claims in this manuscript. Thus, it would be appreciated if data from all 98 patients were provided as supplementary.

Response: We thank the reviewer for the comment because it would have increased the value of this research; we added a supplementary table and modified some sentences in materials and methods and in result sections.

Comment 5: The conclusion is also unclear in the title authors want to assess T-cell Assay after Covid-19, but the conclusion doesn’t provide any insights into how T-cell Assay can be interpreted for Covid-19 vaccination? What was the exact conclusion of the study, and how can it be utilized? Please consider revising it again.

Response: Thanks for the suggestion. We modified our discussion and conclusion, as requested.

Round 2

Reviewer 1 Report

All my questions were properly addressed. 

Author Response

We thank you for the suggestions and for the time spent for us. We revised some 

minor spell and style errors.

Author Response

We thank you for your insightful comments.